

# Potential sources and characteristic occurrence of mobile colistin resistance (*mcr*) gene-harbouring bacteria recovered from the poultry sector: a literature synthesis specific to high-income countries

Madubuike Umunna Anyanwu[1], Ishmael Festus Jaja[2], Charles Odilichukwu R. Okpala[3], Chinwe-Juliana Iwu Jaja[4], James Wabwire Oguttu[5], Kennedy Foinkfu Chah[1] and Vincent Shodeinde Shoyinka[1]

[1] Department of Veterinary Pathology and Microbiology, University of Nigeria, Nsukka, Nsukka, Enugu, Nigeria
[2] Livestock and Pasture Science, University of Fort Hare, Alice, Eastern Cape, South Africa
[3] Department of Functional Food Products Development, Faculty of Biotechnology and Food Science, Wrocław University of Environmental and Life Sciences, Wrocław, Poland
[4] Department of Nursing and Midwifery, Faculty of Medicine and Health Sciences, University of Stellenbosch, Cape Town, Western Cape, South Africa
[5] Department of Agriculture and Animal Health, University of South Africa, Johannesburg, Gauteng, South Africa

Corresponding authors
Ishmael Festus Jaja, ijaja@ufh.ac.za
Charles Odilichukwu R. Okpala, charlesokpala@gmail.com

## ABSTRACT

Understanding the sources, prevalence, phenotypic and genotypic characteristics of *mcr* gene-harbouring bacteria (MGHB) in the poultry sector is crucial to supplement existing information. Through this, the plasmid-mediated colistin resistance (PMCR) could be tackled to improve food safety and reduce public health risks. Therefore, we conducted a literature synthesis of potential sources and characteristic occurrence of MGHB recovered from the poultry sector specific to the high-income countries (HICs). Colistin (COL) is a last-resort antibiotic used for treating deadly infections. For more than 60 years, COL has been used in the poultry sector globally, including the HICs. The emergence and rapid spread of mobile COL resistance (*mcr*) genes threaten the clinical use of COL. Currently, ten *mcr* genes (*mcr*-1 to *mcr*-10) have been described. By horizontal and vertical transfer, the *mcr*-1, *mcr*-2, *mcr*-3, *mcr*-4, *mcr*-5, and *mcr*-9 genes have disseminated in the poultry sector in HICs, thus posing a grave danger to animal and human health, as harboured by *Escherichia coli*, *Klebsiella pneumoniae*, *Salmonella* species, and *Aeromonas* isolates. Conjugative and non-conjugative plasmids are the major backbones for *mcr* in poultry isolates from HICs. The *mcr*-1, *mcr*-3 and *mcr*-9 have been integrated into the chromosome, making them persist among the clones. Transposons, insertion sequences (IS), especially IS*Apl1* located downstream and upstream of *mcr*, and integrons also drive the COL resistance in isolates recovered from the poultry sector in HICs. Genes coding multi-and extensive-drug resistance and virulence factors are often co-carried with *mcr* on chromosome and plasmids in poultry isolates. Transmission of *mcr* to/among poultry strains in HICs is clonally unrestricted. Additionally, the contact with poultry birds, manure, meat/egg, farmer's wears/farm

equipment, consumption of contaminated poultry meat/egg and associated products, and trade of poultry-related products continue to serve as transmission routes of MGHB in HICs. Indeed, the policymakers, especially those involved in antimicrobial resistance and agricultural and poultry sector stakeholders-clinical microbiologists, farmers, veterinarians, occupational health clinicians and related specialists, consumers, and the general public will find this current literature synthesis very useful.

# INTRODUCTION

Antimicrobial resistance (AMR) is a One Health global problem imposing enormous clinical and financial burdens across countries regardless of income level (*Collignon et al., 2018*). It is increasingly recognized that socioeconomic growth and AMR level in various parts of the globe are related (*Malik & Bhattacharyya, 2019*; *Okpala et al., 2021*). Due to better infrastructure, education, *per capita* gross domestic product (GDP), public-health spending, as well as administrative governance, the high-income countries (HICs) are considered low AMR impact regions contributing lesser than the low-and middle-income (LMICs) to the global pool of antimicrobial resistance genes (ARGs) (*Collignon et al., 2018*). There are currently 83 HICs/developed nations, areas, or territories worldwide, with some of these countries sharing borders with LMICs/developing nations (*World Bank, 2018*). Selective pressure for the development and emergence of AMR is prompted by inappropriate antibiotic use (*Serwecinska, 2020*). Still, it is increasingly recognized that the development and spread of resistant organisms are lower in HICs because they have better social and economic infrastructures than the LMICs (*Malik & Bhattacharyya, 2019*). However, the AMR does not respect borders, and ARGs, especially those carried on mobile genetic elements like plasmids, rapidly spread from the point of emergence to other places (*Collignon et al., 2018*).

Colistin (COL) is a highest-priority critically important antibiotic (HP-CIA) used as last-line therapy for deadly infections caused by multi-drug resistant Gram-negative bacilli. Although COL was largely abandoned for human use in the 1970s due to its toxicity, it has been used globally for the recent decades to enhance growth, prophylactic control, and metaphylactic treatment in livestock, especially poultry (*Forde et al., 2018*). COL was barely used in humans from 1970–1994. It is not too long ago that clinicians were forced to use COL to treat highly resistant infections, causing an estimated 700,000 human deaths annually worldwide (*Neill, 2014*). Mutations in chromosomal genes such as *prmAB*, *phoPQ*, *mgrB*, and *crrAB* spread vertically among bacterial clones and were thought to be the only COL resistance mechanism (*Cheng et al., 2016*). Accordingly, there was low interest in bacterial resistance against COL since chromosomal mutations are self-limiting by nature (*Carretto et al., 2018*). But in 2015, plasmid-mediated transmissible

gene, mobile COL resistance (*mcr*-1) was detected in *E. coli* isolates from meats and humans in China (*Liu et al., 2016*), indicating that the clinical usefulness of COL is threatened. This heralded the emergence of a truly pandrug-resistant organism (superbug) (*McGann et al., 2016*). Plasmids are self-replicating DNA independent of chromosomes, and the highly-mobile conjugative plasmids carry resistance genes, which enables the fitness of bacterial survival and rapid spread of resistant organisms (*Mathers, Peirano & Pitout, 2015*). However, non-conjugative plasmids need the machinery from another self-transmissible plasmid to be mobilized (*Smillie et al., 2010*). Scarcely five years ago, have the *mcr*-1 and nine other *mcr* genes (*mcr*-2 to *mcr*-10) with numerous variants been detected in isolates from animals, humans, and the environment in more than 60 countries in six of the seven continents (*Ling et al., 2020*; *Wang et al., 2020*). The *mcr* genes encode transmembrane enzymes, phosphoethanolamine (pEtN) transferases that mediate COL resistance by attaching a pEtN moiety to the lipopolysaccharide (LPS) of lipid A in the outer membrane of Gram-negative bacilli. Additionally, the attachment of pEtN eliminates the negative charges on LPS to which cationic COL/polymyxins have an affinity (*Son et al., 2019*).

World Health Organization (WHO) classification of colistin-resistant organisms (COLROS)/*mcr* gene-harbouring bacteria (MGHB) as "highest-priority" organisms have positioned surveillance researches as essential because the associated disease treatment remains challenging (*World Health Organization (WHO), 2017*). COLROS, especially *mcr* gene-harbouring strain, is potentially resistant to many antimicrobial agents (multi to pandrug-resistant). Thus, its presence in the livestock sector is a monumental threat to food safety and environmental health. As we live in a highly globalized world, contamination of livestock products in developed countries can affect populations in developing countries and *vice versa*, thus jeopardizing global food safety (*Fukuda, 2015*). Foodborne transfer of superbug could result in colonization of handlers, preparers, and consumers of food animals and food animal-related products. Through either untreated or inadequately treated sewage (spills), animal manure, contaminated vectors, as well as fomites, both animals and individuals colonized by superbugs become dissemination sources of these organisms into the environment (*Anyanwu, Jaja & Nwobi, 2020*).

Moreover, a compromise of antimicrobial therapy could occur following the spread of ARGs to other organisms in infected individuals' gut. Unfortunately, diseases associated with superbug are often challenging to treat, resulting in substantial economic losses and fatality. Frighteningly, this could potentially result in an estimated 50 million human deaths annually worldwide by 2050 (*Neill, 2014*). Besides being the cheaper quality protein and nutrient resource, poultry continues to be among the most rapidly global thriving livestock industry. Projected as the most crucial livestock sector by 2025, the poultry industry would increase economies and individual incomes across nations worldwide (*Mottet and Tempio, 2017*). Additionally, the poultry production projections have increased with antibiotics consumption, including the HP-CIAs (*Van Boeckel et al., 2015*).

The potential source and the characteristic occurrence of MGHB, particularly within the poultry sector specific to the HICs should be considered of great importance, reason
being that the state-of-the-art in this research direction seems very scanty to our best knowledge. Understanding the sources, prevalence, phenotypic and genotypic characteristics of MGHB in the poultry sector will help fill this gap, and supplement existing information. Through this, the plasmid-mediated colistin resistance (PMCR) could be tackled in order to improve food safety and reduce public health risks. Based on the above-mentioned, the specific objective of the current work was to perform a literature synthesis of potential sources and characteristic occurrence of MGHB recovered from the poultry sector in HICs. We believe that policy makers (which usually comprise the government and experts), especially those involved in AMR, in addition to agricultural/ poultry sector stakeholders-clinical microbiologists, farmers, veterinarians/veterinary officers, occupational health clinicians and related specialists, consumers/general public will find the output of this literature synthesis very useful.

## SURVEY METHODOLOGY

The first step to carry out this literature synthesis was to put together the research questions. Doing this helped us to discern the specific research objectives to focus on, bearing in mind the intended audience to which this article would be of great benefit. Arriving at a formidable consensus, the next step was that we agreed on the search strategy, which allowed us to flesh out the literature search for prerequisite publications involving the presence of *mcr* genes in isolates from the poultry sector in HICs (as categorized by the World Bank for 2019–2020 (*World Bank, 2018*)).

Essentially, the literature synthesis in this paper was assembled from databases of Pubmed, MEDLINE, EMBASE, ScienceDirect/Scopus, Google Scholar, as well as Web of Knowledge. To have updated distribution of MCR-family gene distribution, screening data from the cross-agency and highly centralized National Database of Antibiotic Resistant Organisms (NDARO) (https://www.ncbi.nlm.nih.gov/pathogens/antimicrobialresistance/) were collected up to August 2021. This shows that we made every effort to capture more recent publications where deemed possible. The key terms and/or text words used for the search include: "poultry birds", "avian", "poultry products", "COL determinants", "transmissible colistinresistancegene," "plasmid-mediated mobile colistin resistance gene", "plasmid-associated COL resistance", "mobile COL resistance", "movable COL resistance genes", "*Enterobacteriaceae*", "*Enterobacterales*", "Gram-negative bacilli", "bacterial isolates", names of different poultry birds and names of each of the HICs, areas, and territories. In all the obtained published papers, the references that were deemed necessary/ relevant were identified.

We also established both inclusion and exclusion criteria, which helped us engage in a rigorous appraisal and brainstorming of all the obtained published papers. Certainly, this assured that the current literature synthesis emerged comprehensive and unbiased. Additionally, those published papers considered relevant to the literature synthesis's content and context were effectively utilized. Further, from the information extracted from the included studies, we documented the surnames of authors, year of paper publication, study site, year of sampling, *mcr* gene(s) assayed, type of sample processed, and the number of isolates genotyped for *mcr*. Other information extracted included the number

**Table 1  Studies reporting plasmid-mediated colistin resistance in poultry sector in high-income Asian countries.**

| Country | Source of isolate | Date of isolation (*mcr* gene assayed) | Number of isolates tested for *mcr* | Identified gene/ variant (Number of organism) | Sequence type and/or phylogroup (Virulence genes) | Plasmid (Associated Insertion sequence) | Additional resistance traits | References |
|---|---|---|---|---|---|---|---|---|
| Taiwan | Poultry birds and meats | 2012–2016 (*mcr*-1) | 122 | *mcr*-1 (35 *E. coli* and 2 (*S.* Typhimurium) | ST38, ST428 and ST1196 | IncHI, IncI2 and IncX4 | $bla_{CMY-2}$, $bla_{CTXM-1}$, $bla_{CMY}$, *sul1* and *floR* | (*Kuo et al., 2016*; *Chiou et al., 2017*; *Liu et al., 2018*) |
| Japan | Chickens and chicken meats | 2000–2018 (*mcr*-1 and *mcr*-5) | 2147 | *mcr*-1 (23 *E. coli*) and *mcr*-3.25 (1 *Aeromonas sobria*) | ST5340, ST48, ST1638, ST1011, ST2690, ST297, ST155, ST117 and ST1684/A | IncI, IncI2 and IncX4 | $bla_{CTX-M-1}$ | (*Kawanishi et al., 2017*; *Nishino et al., 2017*; *Ohsaki et al., 2017*; *Odoi et al., 2021*) |
| South Korea | Chickens and chicken meats | 2000–2018 (*mcr*-1 to *mcr*-8) | 122 | *mcr*-1 (25 *E. coli*) | ST410, ST156, ST10, ST101, ST226, ST162, ST88, ST2732, ST1141, ST162, ST6706 and ST2705 | IncI2 (IS*Apl1*) | $bla_{CTX-M-1}$, $bla_{TEM-1}$, *aph (3′)-Ia*, *aac(3)-IId*, $bla_{CTX-M-65}$, *fosA3*, *aadA1*, $bla_{OXA-10}$, $bla_{CTX-M-65}$, *qnrS1*, *floR*, *cmlA1*, *arr-2*, *tet(A)*, *dfrA14*, *aadA2*, $bla_{TEM-1B}$, *sul3*, *dfrA12*, *sul2*, *tet(M)* and mutations in gyrA and parC | (*Lim et al., 2016*; *Yoon et al., 2018*; *Kim et al., 2019*; *Oh et al., 2020*) |
| Qatar | Chickens | 2016–2017 (*mcr*-1and *mcr*-2) | 90 | *mcr*-1 (14 *E. coli*) | – | – | – | (*Eltai et al., 2018*) |

**Note:**
*mcr*, mobile colistin resistance gene; PMCR, plasmid-mediated colistin resistance; -, no data; Additional resistance traits, resistance factors identified in one *mcr*-positive isolate or pooled factors in more than one *mcr*-positive isolate; Sequence type, Warwick multilocus sequence type of *mcr*-carrying *E. coli* isolates; Virulence genes, genes detected in *mcr*-positive *E. coli* isolates except otherwise stated; Plasmid, plasmid types identified in one or pooled *mcr*-positive isolates; Inc., incompatibility; ESBL, Extended-spectrum β-lactamase; AmpC, Ampicillinase C.

and organism positive for the *mcr* gene, variant of the *mcr* gene detected, the population structure of *mcr*-positive strains, virulence-associated genes, mcr-associated plasmid, and insertion sequence. We also extracted data on additional resistance factors identified in the tested isolate (Refer to Tables 1–3). For emphasis, the crux of the current literature synthesis was to provide a robust snapshot of the potential sources and characteristic occurrence of MGHB recovered from the poultry sector in HICs.

## DISCUSSION OF FINDINGS

### Potential sources of plasmid-mediated colistin resistance in the poultry sector in HICs

COLROS/MGHB could enter the poultry production pyramid in HICs through diverse routes, including day-old chicks contaminated *in ovo* or hatchery, unhygienic livestock feed manufacturers, poultry bird caretakers, and the environment contaminated by vectors (*Moreno et al., 2019*). Extensive use of COL is the primary cause of COL selective pressure (*Touati and Mairi, 2021*). The global poultry industry is reported to have

**Table 2 Studies reporting plasmid mediated colistin resistance in isolates from poultry sector in high-income European countries.**

| Country | Source of isolate | Date of isolation (*mcr* gene assayed) | Number of isolates tested for *mcr* | Identified gene/variant (Number of organism) | Sequence type and/or phylogroup (Virulence genes) | Plasmid (Associated Insertion sequence) | Additional resistance traits | References |
|---|---|---|---|---|---|---|---|---|
| Denmark | Chicken meats | 2012–2014 (*mcr-1*) | 480 | *mcr-1* (5 *E. coli*) | ST359, ST48, ST131, ST1112 and ST2063 | IncI2 and IncX4 | *aadA1*, *aadA5*, *aph (3′)-Ic*, *bla*$_{CMY-2}$, *bla*$_{TEM-1B}$, *dfrA1*, *strA*, *strB*, *sul1*, *sul2*, *tet(B)*, *mph (B)*, *bla*$_{SHV-12}$, *tet (A)*, *aadA2*, *cmlA1*, *sul3*, mutations in GyrA and ParC | (Hasman et al., 2015) |
| United Kingdom | Poultry meats | 2014–2017 (*mcr-1* to *mcr-8*) | 792 | *mcr-1* (2 *Salmonella*) | – | IncHI2, IncI and IncX4 | *bla*$_{TEM-1}$, *tet(A)*, *tet (B)*, *tet(M)*, *catA*, *floR*, *cmlA1* and mutation in GyrA | (Doumith et al., 2016; Sia et al., 2020) |
| Portugal | Turkeys and poultry meats | 2011–2014 (*mcr-1*) | 242 | *mcr-1* (56 *E. coli* and 1 *Salmonella*) | – | – | *bla*$_{TEM-1}$, *bla*$_{SHV-12}$, and *bla*$_{OXA}$-type | (Figueiredo et al., 2016; Manageiro et al., 2017) |
| Spain | Turkeys | 2014 (*mcr-1*) | 1700 | *mcr-1* (3 *E. coli*) | – | – | – | (Quesada et al., 2016) |
| Italy | Poultry birds, meats and eggs | 2012–2015 (*mcr-1* and *mcr-1*) | 652 | *mcr-1* (115 *E. coli* and 16 *Salmonella*) | *E. coli*: ST359, ST156, ST101, ST1086, ST131, ST371, ST1485, ST720, ST345, ST354, ST1266, ST155, ST224, ST744, ST38, ST1431, ST410, ST101, ST69, ST156 and ST10 (A, B1 and B2); *Salmonella*: ST3515, ST52, ST32 and ST45 | IncX4, IncFII, IncFI, IncHI2, IncI and IncN (IS1) | *Int1*, *Int2*, *bla*$_{CTX-M-1}$, *bla*$_{TEM-1}$, *bla*$_{CIT}$, *bla*$_{TEM}$, *bla*$_{SHV}$, *aph(3′)*, *aadA2*, *sul1*, *sul3*, *tet(A)*, *dfrA14*, *bla*$_{TEM-1B}$, *bla*$_{SHV-12}$, *bla*$_{CTX-M-55}$, *bla*$_{TEM-52C}$, *bla*$_{TEM-ID}$, *bla*$_{TEM-1B}$-like, *bla*$_{TEM-1C}$, *bla*$_{TEM-1B}$-like, *sul2*, *aadB*, *strA*, *strB*, *aadA2*, *aadA5*, *aadA1*, *dfrA1*, *qnrS1*, *qnrB19*, *aadA1*-like, *aac(3)-IId*-like, *aac(3)-IIa*, *dfrA14*-like, *tetB*, *tet(M)*-like, *cmlA1*-like, *cmlA1*, *catA1*-like, *mph(A)*, *mph (B)*, *floR* and *floR*-like | (Carnevali et al., 2016; Ghodousi, Bonura & Mammina, 2017; Alba et al., 2018; Apostolakos & Piccirillo, 2018; Carfora et al., 2018) |

| Country | Source of isolate | Date of isolation (*mcr* gene assayed) | Number of isolates tested for *mcr* | Identified gene/variant (Number of organism) | Sequence type and/ or phylogroup (Virulence genes) | Plasmid (Associated Insertion sequence) | Additional resistance traits | References |
|---|---|---|---|---|---|---|---|---|
| Poland | Poultry birds | 2011–2020 (*mcr*-1 to *mcr*-5) | 446 | *mcr*-1.1 (97 *E. coli*) | ST1011, ST93, ST744, ST8979, ST10, ST156, ST1137, ST398, ST48, ST141, ST1611, ST3897, ST37, ST86, ST1167, ST602, ST6286, ST354, ST90, ST1303, ST227, ST191, ST155, ST2566, ST117, ST359, ST2509, ST154, ST753, ST1196, ST189, ST58, ST398, ST949, ST7315, ST919, ST349, ST69, ST88, ST624, ST156, ST162, ST617, ST410, ST5979, ST1851, ST1126, ST4598, ST2001 and ST57 | IncX4, IncFIC (FII), IncFIA, IncHI2, IncFIB, IncHI2A, Col, IncQ1, TrfA and IncB/O/K/Z, | *aac(3)-IIa, aadA1, aac(6′)-Ib-cr, aadA2, aph(3′)-Ia*-like, *aph(3″)-Ib, aph(6)-Id, bla*$_{TEM-1A}$, *catA1*-like, *cmltA1*-like, *dfrA1*-like, *mdf(A)*-like, *sul1, sul2, sul3, tetA, tet(A)*-like, *tet(B), mph(E), lnu(F), bla*$_{CMY-2}$*, bla*$_{TEM-52C}$ *msr(E), qnrB19, lnu(G), qnrS1, bla*$_{TEM-1-like}$*, floR*-like, *apf(6)-Id*-like and *mdf(A)*-like | (*Zając et al., 2019*; *Ćwiek et al., 2021*) |
| Romania | Chickens | 2011–2012 (*mcr*-1) | 92 | *mcr*-1 (17 *E. coli*) | ST744, ST57, ST156 and ST10/A, B1 and D | IncX, IncF and IncI | *aac-3-IIa, aph-3-Ia, strA, strB, sul3, bla*$_{TEM-1}$*, bla*$_{CMY}$ and *tet(A)* | (*Maciuca et al., 2019*) |
| Czech Republic | Poultry birds, meats and environment | 2014 (*mcr*-1 and *mcr*-2) | 67 | *mcr*-1 (110 *E. coli* and 3 *K. pneumoniae*) | ST58, ST354, ST69, ST410, ST1582, ST1589, ST2179, ST7973, ST10, ST93, ST744, ST746, ST756, ST5956, ST38, ST69, ST1079, ST7973, ST1167, ST224, ST1196, ST162 and ST349. Klebsiella: ST147 and ST11 | *IncHI2, IncI2* and *IncX4* (IS1, IS3, IS26, IS1326 and ISApl1) | *bla*$_{TEM}$*, bla*$_{SHV-12}$*, bla*$_{TEM-1}$*, aadA1, aadA2, aph(6)-Id, mef(B), tet(A), tet(R), sul1, sul2, sul3* and chromosomal AmpC mutation | (*Karpíšková et al., 2017*; *Gelbíčová et al., 2019*; *Kubelova et al., 2021*; *Zelendova et al., 2021*) |
| Netherlands | Poultry birds and meats | 2009–2014 (*mcr*-1) | 278 | *mcr*-1 (48 *E. coli*, 13 *Salmonella*, and 2 *K. pneumoniae*) | ST2079, ST1730, ST4512, ST1564, ST10, ST38, ST752, ST209, ST351 and ST117 | IncFIB, IncX4, IncFII, IncHI2, IncHI2A, IncI1, IncI2, IncP and p0111 | *aadA1, aadA2, aadA3, aph(3′)-la, aph(3′)-Ib, aph(3′)-lc, aph(6)-Id, bla*$_{CTX-M-1}$*, bla*$_{TEM-1B}$*, tet(A), lnu(F), cmlA1, catA1, sul1, sul2, sul3, dfrA5* and *bla*$_{SHV-12}$ | (*Kluytmans–van den Bergh et al., 2016*; *Veldman et al., 2016*) |

(Continued)

| Country | Source of isolate | Date of isolation (*mcr* gene assayed) | Number of isolates tested for *mcr* | Identified gene/variant (Number of organism) | Sequence type and/ or phylogroup (Virulence genes) | Plasmid (Associated Insertion sequence) | Additional resistance traits | References |
|---|---|---|---|---|---|---|---|---|
| Belgium | Poultry meats | 2012–2015 (*mcr*-1 and *mcr*-2) | 68 | *mcr*-1 (1 *Salmonella*) | ST3663 | IncX4, IncQ1, IncI1, Col (BS512) and ColpVC | *aadA1*, *bla*$_{TEM-1}$, *sul2* and *dfrA1* | (*Garcia-Graells et al., 2018*) |
| Switzerland | Chicken meats | 2014–2016 (*mcr*-1) | 537 | *mcr*-1 (7 *E. coli*) | ST58, ST1775, ST38, ST226 and ST1049 | IncK2, IncI2 and chromosomal (IS*Apl1*) | *bla*$_{SHV-12}$, *bla*$_{TEM-1}$, *sul2*, *aadA1*, *tet (A)*, *aadA5*, *bla*$_{CTX-M-1}$, *dfrA17*, *bla*$_{TEM-52}$, *bla*$_{CMY-2}$, *InuF*, *erm(42)*, *aad24*, *dfrA1*, *strA* and *strB* | (*Zogg et al., 2016; Donà et al., 2017*) |
| Austria | Poultry birds and meats | 2016 (*mcr*-1) | 4 | *mcr*-1 (3 *E. coli*) | ST10, ST616 and ST43 | IncFIC/FII, IncFIB and IncX4 | *bla*$_{TEM}$ | (*Allerberger et al., 2016*) |
| France | Poultry birds, product and farm boot | 2012–2014 (*mcr*-1) | 275 | *mcr*-1 (23 *E. coli* and 3 *Salmonella*) | – | IncI2 | *bla*$_{CMY-2}$ | (*Perrin-Guyomard et al., 2016; Webb et al., 2017*) |
| Germany | Poultry birds, meats and eggs | 2004–2018 (*mcr*-1 to *mcr*-9) | 1190 | *mcr*-1 (585 *E. coli* and 53 *Salmonella*), *mcr*-2 (10 *Salmonella*), *mcr*-3 (10 *Salmonella*), *mcr*-3.7 (1 *Aeromonas media*), *mcr*-4 (1 *Salmonella mcr*-5 (19 *Salmonella*), and *mcr*-9 (1 *Salmonella* | ST410, ST131 and ST69 | IncX4, IncHI2 and ColE (plasmidal and chromosomal IS*Apl1*) | Tn3, *bla*$_{CTX-M-15}$, *bla*$_{TEM-1}$, *bla*$_{TEM-135}$, *bla*$_{CMY-2}$, *strA*, *strB*, *aadA1*, *aadA2*, *catA1*, *cmlA1*, *sul3*, *tet (M)*, *dfr*, *aph(3')-Ic*, *aac(3)-IIa*, *sul1*, *sul2*, *dfrA1*, *tet(A)*, *catA1*, *floR*, *dfrB8*, *tet (31)*, *ere(A)* and *mph(B)* | (*Ewers et al., 2016; Falgenhauer et al., 2016; Irrgang et al., 2016 (Hornsey et al., 2019; Borowiak et al., 2017; Borowiak et al., 2020*) |

**Note:**

*mcr*, mobile colistin resistance gene; -, no data; Additional resistance traits, resistance factors identified in one *mcr*-positive isolate or pooled factors in more than one *mcr*-positive isolate; Virulence genes, genes from *mcr*-positive *E. coli* isolates except otherwise stated; Sequence type, Warwick multilocus sequence type of *mcr*-harbouring *E. coli* isolates except otherwise stated; Plasmid, plasmid types identified in one or pooled *mcr*-positive isolates; Inc., incompatibility.

consumed 49.01% of total COL sulphate usage in livestock (*Ling et al., 2020*; *Shen et al., 2020a*). In the recent past, European countries were reported to have consumed massive amounts of COL in livestock more than humans (*Webb et al., 2017*). However, in 2016 in Europe, COL was placed in category "B" as a restricted drug whose use in veterinary medicine should be limited to reduce the danger to public health and only be used when there is no other alternative (*Andrade et al., 2020*). COL is poorly absorbed in birds' gastrointestinal tract, and the consequent low bioavailability following oral administration potentially triggers COL selective pressure (*Apostolakos & Piccirillo, 2018*).

**Table 3 Mullticentric studies reporting plasmid-mediated colistin resistance in isolates from poultry meat supply chain in high-income European countries.**

| Country | Source of isolate | Year of isolation (*mcr* gene assayed) | Number of isolates tested for *mcr* | Identified gene/variant (Number of organism) | Sequence type and/or phylogroup | Plasmid (Associated Insertion sequence) | Additional resistance traits | Reference |
|---|---|---|---|---|---|---|---|---|
| Germany, Switzerland, Denmark, Hungary, Italy and Austria | Poultry meats | 2018 (*mcr*-1) | 12 | *mcr*-1 (12 *E. coli*) | ST156, ST3519, ST10, ST650, ST1251, ST58, ST1431, ST355, ST744 and ST1431 | – | – | (*Zogg et al., 2016*) |
| France, Germany, The Nether, Hungary, Spain and UK | Chickens | 2002–2014 (*mcr*-1-like and *mcr*-2) | 119 | *mcr*-1-like (45 *E. coli*) | ST10, ST57, ST165, ST209, ST301, ST373, ST1716, ST752, ST997, ST1011, ST1286, ST1564, ST1842, ST1968, ST2404, ST20, ST85, ST88, ST101, ST156, ST162, ST359, ST763, ST1196, ST1431, ST1730, ST2526, ST2607, ST131, ST141, ST648, ST5204, ST5254, ST5229, ST57676 and ST5848 (A, B1, B2, D and unknown) | – | – | (*El Garch et al., 2018*) |
| Czech Republic, Poland, Hungary, Germany, Slovakia, Austria, Spain, Netherlands, Belgium and Great Britain | Turkey meats | (*mcr*-1 to *mcr*-5) | *mcr*-1d in 118 samples | *mcr*-1(51 *E. coli* and 2 *K. pneumoniae*: | *E. coli*: ST756, ST162, ST2179, ST744, ST10, ST746, ST3956, ST1467, ST1196, ST1081, ST156, ST7973, ST224, ST707, ST1079, ST1589, ST93, ST58, ST1463, ST410, ST1582, ST1011, ST86, ST453, ST1140, ST349, ST69, ST385, ST354 and ST7233; *K. pneumoniae*: ST147, ST3128, ST11 and ST659 | IncX4, IncI2 and IncHI2 | $bla_{SHV-2}$, $bla_{SHV-12}$, *oqxA*, *oqxB*, *qnrS*, *qnrB*, *qnrS1*, *qnrB19*, *aac(6')-Ib-cr*, $bla_{CTX-M-15}$ and $bla_{CTX-M-1}$ | (*Gelbicova et al., 2020*) |
| Czech Republic, Hungary, Poland and Germany | Turkey meats | 2017–2018 (*mcr*-1to *mcr*-5) | *mcr*-1 in 12/17 samples | *mcr*-1 (12 *E. coli* and 1 *K. pneumoniae*) | – | – | – | (*Gelbíčová et al., 2019*) |

**Note:**
*mcr*, mobile colistin resistance gene; -, no data; Additional resistance traits, resistance factors identified in one *mcr*-positive isolate or pooled factors in more than one *mcr*-positive isolate; Sequence type, Warwick multilocus sequence type of *mcr*-harbouring *E. coli* isolates; Plasmid, plasmid types identified in one or pooled *mcr*-positive isolates; Inc., incompatibility.

In the United States of America (USA), COL was approved for use by the US Food and Drug Administration (FDA), although not as a growth promoter in animal feeds (*Ling et al., 2020*). In Canada, COL is not approved for use in veterinary medicine. However, it has been cautiously used orally in managing intestinal diseases in livestock at one time in Canada (*Rhouma et al., 2019*), while COL (although not approved in the continent) is used in food animals at a minimal level in Australia (*Ellem et al., 2017*; *Bean et al., 2020*).

In Asia (with 22 HICs out of 48 countries), South America (with 2 HICs–Uruguay and Chile out of 12 countries) and Africa (with 1 HIC-Seychelles out of 54 countries)

where LMICs predominates more than HICs (*World Bank, 2018*), COL has mostly been used unregulated in livestock; but fortunately, in the recent past, some countries in these regions have banned non-therapeutic COL use in livestock (*Shen et al., 2020a*). Unfortunately, several countries in these continents (South America, Asia, and Africa) are tourist destinations, heavily populated, and exporters of livestock products even to HICs (*Coyne et al., 2019*). China is a developing country but a topmost producer and consumer of COL in livestock, thus serving as a potential source for the dissemination of PMCR due to the exportation of animals and associated products to many countries as well as its high human population frequently travelling all over the world (*Liu et al., 2018*). Fortunately, it has been reported that the prevalence of MGHB in China is reducing due to the enforced ban (since 2017) on non-therapeutic COL use in the Chinese livestock industry (*Shen et al., 2020b*). Nevertheless, some plasmids could capture ARGs even in the absence of selective pressure (*Lopatkin et al., 2016*).

PMCR could also be imported into a HIC through the trade of poultry birds, meat, eggs, and poultry-related products following importation from developed and developing countries (*Grami et al., 2016*). Meat contamination is easy, especially in LMICs, due to unhygienic animal slaughter practices and an unsanitary slaughterhouse environment (*Jaja et al., 2020*). The lack of pre-slaughter assessment of COL usage in food animals and lack of post-slaughter assessment of meats for the presence of COLROS/*mcr* genes makes PMCR rapidly spread from one place to another through meat trade. Visitation to areas with high PMCR prevalence is a putative risk for colonization by MGHB. Handling and consumption of contaminated food (especially animal-related food) and liquid, direct or indirect contact with colonized/infected animal, person, or contaminated fomites are potential sources for acquiring MGHB in areas of high PMCR endemicity. Travelers could transport MGHB from places of visitation back to their home countries/household, potentially result in community transmission of PMCR in countries with low PMCR prevalence (*Frost et al., 2019*).

## Characteristic occurrence of mcr-carrying isolates in the poultry industry across high-income countries (HICs)

### Asia

The high population density, increasing economies of many nations, high disease burden, and increasing livestock intensification are factors that potentially facilitate the development of AMR in Asia and its dissemination from this region to other parts of the globe (*Coyne et al., 2019*). Eleven studies investigated PMCR in 4596 isolates from the poultry sector in four HICs (out of 22 HICs/territories) in Asia (Table 1). Ninety-nine isolates (97 *E. coli* and 2 *Salmonella enterica* subspecies enterica serovar Typhimurium (*S.* Typhimurium) and one *Aeromonas sobria*) were reported to harbour the *mcr*-1 and *mcr*-3 gene, respectively.

### -Eastern Asia

The poultry sector in Taiwan has been reported as a reservoir for *mcr*-1-habouring *Enterobacteriaceae* (*Kuo et al., 2016*; *Chiou et al., 2017*; *Liu et al., 2018*). Thirty-seven

isolates (35 *E. coli* and 2 *S.* Typhimurium) carrying *mcr*-1 on ~60 kb IncI2, ~43 kb IncX4, and 60–300 kb IncHI2 plasmids and chromosome (in eight *E. coli* strains) were detected among 122 *E. coli* (30.3%) recovered 2012–2016 from chickens and samples of chicken meats (*Kuo et al., 2016*; *Chiou et al., 2017*; *Liu et al., 2018*), indicating that diverse promiscuous plasmids have widely spread *mcr*-1 among *Enterobacteriaceae* circulating in Taiwanese poultry meat sector since at least 8 years ago. It further indicates that *mcr*-1 was transferred vertically to progenies, thus persisting among *E. coli* clones in Asia. IS*Apl1* was upstream of *mcr*-1 in the plasmids of some *mcr*-1-positive isolates, which were heterogeneous belonging to three STs (*Kuo et al., 2016*) (Table 1) dominated by high-risk (HiR) zoonotic pandemic extraintestinal pathogenic *E. coli* (ExPEC) clone ST38 (*Manges et al., 2019*). Thus, in Taiwan, diverse genetic elements (plasmids and transposons) drive the acquisition/transfer of *mcr*-1 among commensal and virulent *E. coli* clones. Some of the *mcr*-1-positive *E. coli* isolates (recovered 2013) were producers of extended-spectrum β-lactamases (ESBL) (*Kuo et al., 2016*) while the *mcr*-1-positive salmonellae were multidrug-resistant Ampicillinase C (AmpC) producers (*Chiou et al., 2017*), suggesting that potentially pandrug-resistant *Enterobacteriaceae* coproducing MCR-1 and ESBL/AmpC has been present Taiwan since at least 7 years ago, thus posing a huge danger to animal and public health. Fortunately, there is now a ban on non-therapeutic COL use in the Taiwanese livestock sector (*Liu et al., 2018*).

Japanese poultry sector has also been noted as a reservoir for MGHB (*Kawanishi et al., 2017*; *Nishino et al., 2017*; *Ohsaki et al., 2017*). A total of thirty-seven *E. coli* carrying *mcr*-1 (*mcr*-1.21 in one strain) on ~60 kb IncI and 30 kb IncX4 plasmids and one chromosomal *mcr*-3.25-positive *Aeromonas sobria*) were detected among 2147 *Enterobacterales* (1.8%) isolated 2000–2018 from healthy broilers, retail chicken meats, and chicken meats sourced locally and imported from Brazil into Japan (*Kawanishi et al., 2017*; *Nishino et al., 2017*; *Ohsaki et al., 2017*; *Odoi et al., 2021*). This points to the fact that both local and external poultry meat supply chains are very potential routes through which PMCR might have been disseminated in Japan, albeit at a low prevalence. It also suggested that in Japan, COLROS possibly cross-contaminate poultry meat from the chicken meat handlers and/or fomites during the processing (*Figueiredo et al., 2016*). The *mcr*-1-positive isolates were extensively diversified belonging to nine STs (*Nishino et al., 2017*; *Ohsaki et al., 2017*) (Table 1), including HiR pandemic ExPEC clone ST117 (*Manges et al., 2019*), indicating that diverse commensal and virulent *E. coli* clones are spreading *mcr*-1 in Japan. Lamentably, the *mcr*-1-IncI conjugated with a recipient organism (*Kawanishi et al., 2017*), indicative that the isolates could quickly transfer *mcr*-1 to other organisms. Although most of the *mcr*-1-harbouring strains from the imported meats were ESBL-producers exhibiting multidrug resistance (*Nishino et al., 2017*; *Ohsaki et al., 2017*), some of them were susceptible to all the 13 antimicrobial agents tested (*Nishino et al., 2017*). This suggests that organisms coproducing MCR-1 and ESBL were possibly imported into Japan and that *mcr* gene does not necessarily confer multi-, extensive or pandrug resistance. This highlights the need to conduct antimicrobial susceptibility testing (AST) prior to the prescription/administration of antibiotics. Howbeit, Japan imports food from many countries, which predisposes it to the imported AMR. But encouragingly, risk management for COL in

livestock animals, including enhanced monitoring of the antimicrobial-resistant bacteria, restriction of COL to a second choice drug status, and revocation of its designation as a feed additive, is being promoted in Japan (*Ling et al., 2020*).

Evidence has shown that the South Korean poultry sector constitutes a reservoir for COLROS (*Lim et al., 2016*; *Yoon et al., 2018*; *Kim et al., 2019*; *Oh et al., 2020*). Twenty-five strains carrying *mcr*-1 on IncI2 plasmid were detected among 122 *E. coli* (20.5%) isolated 2000–2018 from healthy/diseased chickens, and samples of chicken meats collected from processing facilities (*Lim et al., 2016*; *Yoon et al., 2018*; *Kim et al., 2019*; *Oh et al., 2020*), indicating that *mcr*-1 has been widely spread by IncI in South Korean poultry industry since at least in 2013. IS*Apl1* flanked *mcr*-1 upstream in one of the strains (*Yoon et al., 2018*), meaning that diverse genetic elements (plasmids and transposons) facilitate the acquisition/spread of COL resistance in South Korea. The *mcr*-1-positive isolates extensively diversified belonging to 12 STs (*Lim et al., 2016*; *Oh et al., 2020*), including HiR pandemic ExPEC clones ST410, ST10, and ST88 (*Manges, 2016*), and there were 20 extra resistance genes (including ESBL, fosfomycin and plasmid-mediated fluoroquinolones resistance (PMQR) genes) belonging to eight antimicrobial families in them (Table 1). This suggests that the transmission of *mcr*-1 among *E. coli* strains in South Korea is non-clonal. These strains are spreading resistance against last resort antimicrobial agents in the country. Unfortunately, these organisms could rapidly transfer multi to pandrug resistance to other organisms, having transferred *mcr*-1 to a recipient organism at a very high frequency of $10^{-2}$ to $10^{-6}$ (*Kim et al., 2019*). However, there was no other resistance gene in one of the *mcr*-1-positive isolates (*Yoon et al., 2018*), implying that *mcr*-1 is acquired without selective pressure being exerted by non-COL antimicrobial agents.

Since COL has been used for a long time in the South Korean livestock sector with annual consumption of 6–16.3 tons in 2005–2015 (*Kim et al., 2019*), the selective pressure for the acquisition of *mcr* genes very likely originated from the livestock sector from where PMCR disseminated into the human-environmental ecosystem (*Yoon et al., 2018*), possibly by contact with/consumption of livestock/related products in the country. Nevertheless, mutations in chromosomally-encoded *pmrB*, *phoP*, *phoQ*, *mgrB*, and *pmrD* have also been detected in isolates from food animals in South Korea (*Kim et al., 2019*), further supporting that non-*mcr* mechanism also mediate COL resistance in Eastern Asia.

### -Southeastern Asia

The presence of *mcr*-1.8 in *E. coli* isolates of poultry origin has been documented in Brunei (GenBank Accession Number: NG_054697). This means that *mcr*-1 is circulating in the only HIC in South East Asia (SEA) (*World Bank, 2018*), a region made up of countries that are heavy producers/exporters of poultry and aquaculture products (*Coyne et al., 2019*).

### Middle East

Among the six HICs (Oman, Qatar, Kuwait, United Arab Emirates, Israel, and Saudi Arabia) in the Middle East (*World Bank, 2018*), there appears to be only one study from

Qatar that reported PMCR in the poultry sector. In the study, 14 *mcr*-1-carrying isolates were detected among 90 MDR *E. coli* (15.6%) isolated 2016-2017 from cloacal swabs of broilers (*Eltai et al., 2018*). This buttresses that *mcr*-1-harbouring *E. coli* strains colonize a sizeable percentage of chickens in Qatar, posing a threat to public health. Regrettably, *mcr*-1-carrying *E. coli* has already been disseminated into Qatar's human population (*Forde et al., 2018*), possibly from the livestock sector.

*Europe*

The ban on the use of many antimicrobials as feed additives in Europe in 1999 and 2006 resulted in increased use of many antibiotics, especially tetracycline and COL for therapeutic purposes in the continent (*Casewell et al., 2003*). In the recent past, about 28.7% of COL produced in China ended up in Europe (*Liu et al., 2018*). A sum of 545.2 tonnes of active polymyxin ingredients (including COL and polymyxin B) was used in 2012, primarily in the poultry and swine sectors in 22 European countries (*Webb et al., 2017*). In 2013, COL was recommended only to treat animal diseases but not for metaphylaxis in livestock (*Miguela-Villoldo et al., 2019*). However, COL has been used for prophylactic control of intestinal diseases in livestock in Europe, with polymyxin being the fifth most commonly sold antimicrobial class in 2013 (*Webb et al., 2017*). So, the use of COL in the livestock sector exerted selective pressure for PMCR in this continent. COL determinant originating from one country in Europe could easily spread to another due to the free trade movement allowing the cross-border transfer of livestock and associated products and individuals (*Lepape et al., 2020*). In 2016, member States in Europe were called to achieve a 65% reduction of COL sales by 2020 (*Lepape et al., 2020*). Understanding the occurrence and characteristics of MGHB in the poultry meat supply chain in European countries will help improve strategies to curb the spread of PMCR. Thirty-eight publications probed PMCR in 10,696 isolates of poultry origin in European HICs (Table 3). The *mcr*-1 variants were detected in 1,241 strains (1144 *E. coli*, 93 *Salmonella*, one *Aeromonas* and three *Klebsiella*), *mcr*-2, *mcr*-3, *mcr*-4, *mcr*-5 and *mcr*-9 in 10, 10, 1, 19 and 1*Salmonella*, respectively. Three of the publications detected *mcr* genes by direct sample testing.

**-Northern Europe**

The poultry meat supply chain in northern Europe has been noted as a reservoir of the *mcr*-1 gene (*Hasman et al., 2015*; *Doumith et al., 2016*; *Sia et al., 2020*). A total of five AmpC-/ESBL-producing strains carrying *mcr*-1 on IncI2 plasmid (in four strains) were detected among 480 *E. coli* (1%) isolated from samples of chicken meats sourced locally and imported into Denmark during 2012–2014 (*Hasman et al., 2015*). There were 17 other resistance genes in eight different antimicrobial families (including ESBL and AmpC genes and chromosomal resistance genes) in the extensively diverse strains belonging to five STs, including HiR pandemic ExPEC clone ST131 (*Manges et al., 2019*). This indicates that there has been a low circulation of commensal and virulent *E. coli* strains coproducing MCR-1 and ESBL/AmpC in the Nordic region for at least 8 years ago. The ST131 *E. coli* is known to have a broad-host-range and is mostly associated with

difficult-to-treat urinary tract and bloodstream infections in humans related to their capacity to carry multi-resistance virulence genes without fitness cost (*Manges et al., 2019*). Thus, its presence in the poultry industry poses a significant danger to public health, especially to handlers and consumers of these meats. Unhappily, MGHB has already disseminated into the human-environmental ecosystem in Denmark, though later than in the poultry sector (*Zurfluh et al., 2016*).

In the United Kingdom, the *mcr*-1 carried on IncHI2 plasmid was detected in two *Salmonella* Paratyphi B var Java PT Colindale isolated from poultry meat imported (in 2014) into England/Wales from Europe (*Doumith et al., 2016*), suggesting that *mcr*-1-carrying organisms might have been imported into the UK poultry meat supply chain since at least in about 2014. Recently, *mcr*-1 was also detected in a *Salmonella enterica* serovar Java isolated from poultry meat equally sampled in 2014 (*Sia et al., 2020*), further supporting that *mcr*-1 has been present in the UK's poultry meat supply chain since at least 6 years ago. Because COL is used more in patients in the UK than in other European countries (*Catry et al., 2015*; *Doumith et al., 2016*) and *mcr*-1 has been detected earlier in humans in the UK (*Public Health England (PHE), 2015*), the *mcr*-1 in poultry could be of anthropogenic origin. Other COL resistance determinants such as *mcr*-2-and *mcr*-3 were detected in salmonellae recovered from humans and the environment. It also suggests that the protocol used for treating anthropogenic/agricultural sewages/wastes cannot destroy *mcr* genes, thereby allowing the genes to escape into the environment. This highlights the need to improve sewage treatment protocols to ensure the complete elimination of ARGs in waste before releasing them into the environment (*Anyanwu, Jaja & Nwobi, 2020*). There is also a high possibility of farm-to-plate transmission since the COL resistance determinants were also detected in the study's food samples. This further highlights the importance of cooking food properly and the maintenance of strict hygiene by food vendors. Diverse MGEs, including plasmids, insertions sequences, and transposons, was also found in the strains (*Sia et al., 2020*), suggesting that these are the drivers of COL resistance in the UK.

### -Southern Europe

The presence of MGHB in the poultry sector in Southern Europe has also been documented (*Carnevali et al., 2016*; *Figueiredo et al., 2016*; *Quesada et al., 2016*; *Zogg et al., 2016*; *Manageiro et al., 2017*; *Alba et al., 2018*; *Clemente et al., 2019*; *Apostolakos & Piccirillo, 2018*; *Koutsianos et al., 2021*). In Portugal, 57 (23.6%) *mcr*-1-harbouring strains (56 *E. coli* and 1 *S.* Typhimurium) were detected among 242 *Enterobacteriaceae* recovered in 2011/2014 from caeca of poultry birds and meats (*Figueiredo et al., 2016*; *Manageiro et al., 2017*; *Clemente et al., 2019*), suggesting a high prevalence of *mcr*-1 among *Enterobacteriaceae* in the Portuguese poultry meat sector since the year 2011. It could not be categorically stated that MGHB colonized poultry birds in Europe at that period because the exact geographical origin of the meats from which some of the *mcr*-1-positive *E. coli* were isolated in 2011 could not be ascertained. Also, there was a high possibility of cross-contamination during meat processing (*Figueiredo et al., 2016*). Nevertheless, the *Salmonella* transferred *mcr*-1 to a recipient organism at a frequency of ~$10^{-4}$, meaning that

the strain could rapidly transfer *mcr*-1 to other microorganisms, thus posing a risk to public health. There were also β-lactam resistance genes (including ESBL gene) in the *mcr*-1-positive *E. coli* isolates (*Manageiro et al., 2017*; *Clemente et al., 2019*) (Table 2), indicating that *E. coli* coproducing *mcr*-1 and ESBL has been present in Portugal since at least around 2014. The β-lactam could co-select for COL resistance. Slaughtered birds are a potential source for contamination of slaughterhouse environment/personnel, and the manure from them used as farm fertilizers could introduce MGHB into the environment. Unfortunately, *mcr*-1-positive *E. coli* has also been isolated from humans and botanical ecosystems in Portugal (*Anyanwu, Jaja & Nwobi, 2020*). In Greece, one MDR *mcr*-1-positive strain was detected among 150 *E. coli* (0.7%) isolated from commercial layers with colibacillosis (*Koutsianos et al., 2021*). This specific finding would suggest that *mcr*-1 would be circulating probably at low rate in the Greece poultry sector.

In Spain, three faecal *E. coli* (isolated 2014) carrying *mcr*-1 on 50–70 kb IncI plasmid were isolated from 1,700 turkeys (*Quesada et al., 2016*), indicating that IncI has spread *mcr*-1among *E. coli* strains colonizing poultry birds in Spain since at least 6 years ago. Conjugation was positive at a frequency of $2.6 \times 10^{-1}$ to $5.1 \times 10^{-2}$, meaning that the organisms could rapidly transfer the *mcr*-1 to other organisms. COL selective pressure in Spain is likely due to use in food animals as reported consumption of COL was > 100 tons with more than 20 mg/kg of animal biomass of COL used in 2013 (*Webb et al., 2017*). Contact with livestock, use of insufficiently-treated animal manure in farmlands and/or discharge of these manures into water bodies might have facilitated the spread of PMCR into the human-environmental ecosystems in Spain (*Anyanwu, Jaja & Nwobi, 2020*). PMCR has been reported in Italy's poultry sector, the second member of the European Union with the most extensive use of polymyxins in veterinary medicine (*Carnevali et al., 2016*). Atotal of one hundred and thirty-one strains (115 *E. coli* and 16 salmonellae) possessing *mcr*-1.1, *mcr*-1.2, and a new *mcr*-1 variant, named *mcr*-1.13 were detected among 652 MDR *Enterobacteriaceae* (20.1%) isolated from poultry birds, meats, and eggs (*Carnevali et al., 2016*; *Zogg et al., 2016*; *Ghodousi, Bonura & Mammina, 2017*; *Alba et al., 2018*; *Apostolakos & Piccirillo, 2018*; *Carfora et al., 2018*). This discovery suggests that *mcr*-1 variants are widely spread among *Enterobacteriaceae* in the Italian poultry sector. In the *E. coli* isolates, *mcr*-1 was on various plasmids, mostly IncX4 (Table 2), IS66 and IS110 were upstream and downstream in one strain, and class 1 and 2 integrons were also in some of them (*Alba et al., 2018*). These strains were extensively diverse, belonging to 23 STs (dominated by ST10) (*Zogg et al., 2016*; *Alba et al., 2018*) (Table 2), including zoonotic HiR virulent pandemic ExPEC clones ST10, ST131, ST38, ST69, ST410, and ST354 (*Manges et al., 2019*) and haboured 31 extra resistance genes (including ESBL and AmpC genes) belonging to eight antimicrobial families (*Zogg et al., 2016*; *Ghodousi, Bonura & Mammina, 2017*; *Alba et al., 2018*; *Apostolakos & Piccirillo, 2018*). Some of the *mcr*-1-harbouring salmonellae were recovered in 2012 (*Carnevali et al., 2016*), and they contained 11 other resistance genes (including ESBL gene) in six antimicrobial classes and many virulence/fitness-enhancing genes in some others (*Carfora et al., 2018*; *Alba et al., 2018*) (Table 2). These findings indicate that virulent *Enterobacteriaceae* coproducing MCR-1 and ESBL has been present in Italy since at least 8

years ago. Those diverse genetic elements (plasmids, transposons, and integrons) facilitate the acquisition/spread of multi to extensive resistance among *Enterobacteriaceae* in the Italian poultry industry. Unfortunately, the *mcr*-1 could be transferred to other organisms from the recipient organism (*Apostolakos & Piccirillo, 2018*). Regrettably, the dissemination of PMCR into the Italian human-environmental ecosystems has already been documented (*Anyanwu, Jaja & Nwobi, 2020*). This further highlights the need to reduce all antimicrobial agents at the primary level of livestock production. The reduction will mitigate the effects of complex mechanisms of co-selection and multidrug resistance in "Consumer Protection" and "One Health" perspective (*Alba et al., 2018*).

Remarkable, some of the *mcr*-1-positive strains exhibited wild-type COL MIC (<2 μg/mL) (*Ghodousi, Bonura & Mammina, 2017*; *Apostolakos & Piccirillo, 2018*), suggesting that the magnitude of PMCR in an ecological niche could be underestimated by testing only isolates within the recommended COL ecological value (ECV) (≥2 μg/mL). This means that the accurate prevalence of *mcr* gene can solely be determined by screening all isolates from an ecological niche for the *mcr* gene irrespective of the organisms' COL MIC (*Apostolakos & Piccirillo, 2018*). Nevertheless, direct sample testing before isolation in COL resistance surveillance remains the best approach for determining the magnitude of *mcr* gene in a potential reservoir.

### -Eastern Europe

The poultry industry in Eastern Europe has been reported to be a reservoir for MGHB (*Karpíšková et al., 2017*; *Zając et al., 2019*; *Gelbíčová et al., 2019*; *Gelbicova et al., 2020*, *Ćwiek et al., 2021*). In Poland, 97 isolates carrying *mcr*-1.1 on diverse plasmids (IncX4, IncFI, IncHI2, IncQ1, TrfA, IncB/O/K/Z, and many others) were detected among 128 faecal COL-resistant *E. coli* (62.5%) recovered from poultry birds during 2011–2020 (*Zając et al., 2019*; *Ćwiek et al., 2021*), suggesting that diverse promiscuous plasmids evolved *mcr*-1 in Poland since at least in 2011. There was an increase in *mcr*-1-carrying strains from 1.1% in 2011 to 11.6% in 2016, suggesting possible increasing use of COL in breeder farms in countries where day-old chicks are imported into Poland (*Zając et al., 2019*). A total of one of the *mcr*-1-positive isolates had a mutation in the chromosomal *pmrB* gene, further indicating that chromosomal and plasmid-mediated mechanisms simultaneously mediate COL resistance in *E. coli* strains of avian origin. There were 29 additional resistance genes (including ESBL, AmpC, and PMQR genes) in the *mcr*-1-harbouring isolates, which were extensively diverse belonging to 50 STs (*Zając et al., 2019*; *Ćwiek et al., 2021*) (Table 2), including HiR pandemic ExPEC clones ST167, ST38, ST117, ST58, ST69, ST88, ST10 and ST410 (*Manges et al., 2019*). This diversity proves that various plasmids resulted in a diverse range of clones carrying *mcr*-1 and genes coding against last-resort antimicrobials in Eastern Europe. Some of the *mcr*-1-harbouring isolates also exhibited resistance against tigecycline, and some haboured virulence-associated genes, including *astA* gene encoding heat-stable enterotoxin-1 associated with diarrhoea in humans (*Anyanwu, Jaja & Nwobi, 2020*). *mcr*-1-carrying *E. coli* has been associated with diarrhoea in an individual who possibly contacted livestock in Poland (*Izdebski et al., 2016*). Since tigecycline is a last-line agent for treating COL-resistant infections, infection

by a tigecycline-resistant organism could result in death. Therefore, increased surveillance of PMCR and tigecycline resistance in livestock/animal products is urgently warranted to mitigate an impending global health catastrophe.

In Romania, 17 *E. coli* strains carrying *mcr*-1 on IncF, IncX, and IncI plasmids were detected among 92 AmpC-producing *Enterobacterales* (18.5%) isolated in 2011/2012 from caeca of 92 slaughtered chickens (*Maciuca et al., 2019*). This illustrates that diverse plasmids have widely spread *mcr*-1 in MDR organisms colonizing poultry birds in Romania since 2011. The *mcr*-1 was transferred to a recipient organism, and a composite transposon (Tn*6330*) and ISApl1 were upstream and downstream of *mcr*-1 in the isolates. This means that the *mcr*-1 was acquired horizontally and transferrable to other organisms. Eight additional resistance genes belonging to five antimicrobial families were present in theisolates which were heterogenous belonging to phylogroup B/A/D and four STs (dominated by ST744 and ST57) (Table 2), including pandemic HiR pandemic ExPEC clones ST10 and ST57 (*Manges et al., 2019*), indicating that Romanian poultry sector constitutes a potential reservoir for highly-virulent *E. coli* clones thus posing a threat to public health. Fortunately, none of the poultry abattoir workers sampled in Romania harboured organisms carrying *mcr*-1 or *mcr*-2 (*Maciuca et al., 2019*). This is possibly due to hygienic slaughter techniques employed in the sampled slaughterhouses.

In the Czech Republic, 113 strains (110 *E. coli* and 3 *K. pneumoniae*) carrying *mcr*-1 were isolated from poultry meats sourced locally and imported into the country from Poland, Germany, and Brazil (*Karpíšková et al., 2017*; *Gelbíčová et al., 2019*; *Gelbicova et al., 2020*; *Kubelova et al., 2021*; *Zelendova et al., 2021*), suggesting that PMCR in Czechia poultry meat supply chain originate locally. Hence meat trade is a route for global dissemination of PMCR. The *mcr*-1 was on 34–45 kb IncX4, 55–60 kb IncI2, and 250–260 kb IncHI2 plasmids (with IncX4 predominating) in some of the isolates (*Gelbíčová et al., 2019*; *Zelendova et al., 2021*), indicating that IncI and IncH plasmids are the commonest drivers of PMCR resistance in the poultry environment in the countries from where the meats originated. There were nine other resistance genes (including ESBL and PMQR) in the strains (*Gelbíčová et al., 2019*; *Zelendova et al., 2021*) (Table 3), further supporting that poultry meat remains a potential source for spreading multi-to extensively drug-resistant organisms that would pose severe risk to public health. Conjugation was positive, and the microorganisms were extensively diverse, with the *E. coli* isolates belonging to 30 STs, including HiR pandemic ExPEC clones ST10, ST58, ST354, ST69, and 410 (*Gelbíčová et al., 2019*; *Manges et al., 2019*; *Kubelova et al., 2021*; *Zelendova et al., 2021*). In comparison, the *K. pneumoniae* isolates belonged to four STs (*Gelbíčová et al., 2019*; *Zelendova et al., 2021*) (Table 3). This suggests the exchange of *mcr*-1 between diverse clones of enterobacteria, thus increasing public health risks. It also implies that several enterobacterial lineages are carrying *mcr*-1 in European Economic Area (EEA). Hence the need for post-slaughter/pre-exportation assessment of meats for the presence of MGHB to minimize the risk posed by contaminated meats to public health. Since meats are not cooked for a long duration by many Europeans as done by people from some other parts of the world, MGHB in meats consumed in Europe can easily colonize the consumers, potentially jeopardizing antimicrobial therapy.

### -Western Europe

The poultry production chain in Western Europe has been reported to be a reservoir for MGHB (*Allerberger et al., 2016*; *Ewers et al., 2016*; *Falgenhauer et al., 2016*; *Irrgang et al., 2016*; *Kluytmans–van den Bergh et al., 2016*; *Veldman et al., 2016*; *Zogg et al., 2016*; *Perrin-Guyomard et al., 2016*; *Borowiak et al., 2017*; *Donà et al., 2017*; *Schrauwen et al., 2017*; *El Garch et al., 2018*; *Eichhorn et al., 2018*; *Garcia-Graells et al., 2018*; *Webb et al., 2017*; *Hornsey et al., 2019*; *Borowiak et al., 2020*). In the Netherlands, 63 strains (48 *E. coli*, 13 salmonellae, and two *K. pneumoniae*) carrying *mcr*-1 were detected among 278 *Enterobacteriaceae* (22.7%) isolated 2002–2014 from chickens and samples of poultry meat sourced domestically and imported from other European countries (*Veldman et al., 2016*; *Kluytmans–van den Bergh et al., 2016*; *Schrauwen et al., 2017*; *El Garch et al., 2018*). The *mcr*-1 was carried on diverse plasmids, especially 225–290 kb IncHI2 and 20–60 kb IncX4 in some isolates (Table 2), while it was flanked upstream by IS*Apl1* in others (*Veldman et al., 2016*). These findings suggest a wide circulation of *mcr*-1 among *Enterobacteriaceae* in the Dutch poultry meat supply chain. Diverse genetic elements (plasmids and transposons) have been driving the spread of *mcr*-1 in the Netherlands since at least more than a decade ago. It further proves that the trade of poultry meat is a route for the dissemination of MGHB across the Eurozone. Eighteen additional resistance genes belonging to seven antimicrobial families were present in some of the *mcr*-1-positive *E. coli* isolates (*Kluytmans–van den Bergh et al., 2016*; *Veldman et al., 2016*), and the genomes of most of these strains conjugated with that of a recipient organism (*Veldman et al., 2016*). Moreover, the *mcr*-1-positive *E. coli* isolates were extensively diverse belonging to ten STs (*Kluytmans–van den Bergh et al., 2016*; *Veldman et al., 2016*), including zoonotic HiR pandemic ExPEC clones ST10 and ST38 (*Manges et al., 2019*). Thus, diverse commensal and pathogenic *E. coli* clones in the Dutch poultry industry could easily transfer multi-resistance to other organisms, thereby posing a massive threat to public health. Unfortunately, *mcr*-1 has disseminated into the human ecosystem in the Netherlands, possibly from the livestock sector or due to the frequent use of COL for selective gut decontamination in intensive care unit and stem cell transplantation patients (*Nijhuis et al., 2016*; *Terveer et al., 2017*).

A total of thirty-four of the *mcr*-1-harbouring isolates (32 *E. coli* and 2 *K. pnuemoniae*) were recovered from 53 *mcr*-1-positive chicken meat samples (*Schrauwen et al., 2017*), further indicating that by only isolation, the magnitude of COL resistance in food samples could be underestimated. Therefore, direct sample testing followed by isolation is a better approach for surveillance of PMCR in animal origin food. Furthermore, the culture approach showed that the *mcr*-1-carrying strains were susceptible to cephalosporins, carbapenems, and aminoglycosides. This means that the magnitude of PMCR in an ecological niche could also be underestimated if the presence of *mcr* gene is assessed in isolates recovered by selection approach using non-COL antibiotics (*Schrauwen et al., 2017*). It is worthy to note that in NDARO database, sequence of *mcr*-4.6 and other resistance genes were detected in *E. coli* isolated 2019 from Dutch poultry, meaning that *mcr*-4 is present in the Netherlands.
In Belgium, an ST3663 MDR strain (isolated 2015) carrying *mcr*-1 on IncX4 plasmid and four other resistance genes in four antimicrobial families (Table 2) on other various plasmids was detected among 68 COL-resistant salmonellae (1.5%) recovered from chickens/poultry meats (*Garcia-Graells et al., 2018*). This means there has been a low circulation of *mcr*-1-carrying *Salmonella* in the Belgian poultry meat production chain since at least 5 years ago, posing a risk to public health.

In Switzerland, seven strains carrying *mcr*-1 were detected among 537 *E. coli* (1.3%) isolated from chicken meats imported into Switzerland from Germany during 2012–2016 (*Donà et al., 2017*; *Zogg et al., 2016*). The low circulation of *mcr*-1 in the Swiss poultry meat sector is associated with importing poultry meat from Germany. The *mcr*-1-positive isolates were extensively diverse, belonging to five STs (*Donà et al., 2017*), including zoonotic HiR pandemic ExPEC clones ST58 and ST38, which dominated (*Manges et al., 2019*) and16 other resistance genes (including AmpC and ESBL genes) in seven different antimicrobial families were also in the isolates (*Donà et al., 2017*). This finding suggests that organisms coproducing MCR-1 and ESBL might have been imported into Switzerland at least 8 years ago and further supporting that meat is a potential vehicle for disseminating diverse *E. coli* clones capable of causing hard-to-treat diseases. IS*Apl1* was upstream of *mcr*-1 in 65 kb IncI2 plasmid (in four isolates), and a new 100 kb IncK2 plasmid (in two isolates) (*Donà et al., 2017*) and chromosome (*Donà et al., 2017*; *Zogg et al., 2016*) of the organisms and it was transferred to a recipient organism at a very high frequency of $2.6 \times 10^{-4}$ to $6.3 \times 10^{-6}$. This means that IS*Apl1* translocates*mcr*-1 into the chromosome of *E. coli* enabling its maintenance/persistence in the poultry meat supply chain by vertical transmission of the gene to progenies among diverse clones and that these strains could rapidly transfer/acquire *mcr*-1 to/from other organisms and thus posing a worrisome threat to public health. It further shows that poultry meat is a vehicle for dispersing novel *mcr*-associated genetic elements. Sadly, *mcr*-1 has disseminated into Switzerland's human-environmental ecosystem though at a low level (*Zogg et al., 2016*). The low incidence could be due to either the non-use of COL in treating community-acquired infection in Switzerland (*Liassine et al., 2016*; *Büchler et al., 2018*) or the limitation of the methods used to screen the isolates (*Liassine et al., 2016*). However, since COL is used in treating sick animals in Switzerland and given the continued importation of poultry meat (into Switzerland) and potential travel to countries with high endemicity, the human carriage of MGHB in the country could rise if unchecked (*Büchler et al., 2018*). This further highlights the need for post-slaughter assessment of meats for *mcr* even before exportation/importation, especially in Europe, where there is a tendency to cook meat for a short duration. Nevertheless, it is worthy to note that *mcr*-1 was not detected in 40 chicken meat samples collected from Switzerland in 2016 (*Zogg et al., 2016*).

In Austria, three (1.8%) strains of ST10, ST43, and ST616 carrying *mcr*-1 on diverse plasmids (IncFIC/FII, IncFIB, and IncX4) were isolated from caeca of 164 poultry birds and chicken meats of domestic and Italian origin collected in 2016 from slaughterhouses and retail outlets (*Allerberger et al., 2016*). Hence, various promiscuous plasmids spread *mcr*-1 but at a low prevalence in Austria since at least 4 years ago, and that PMCR

might also have been imported into Austria through meat trade. It equally suggested that slaughterhouse and retail points are critical points for contamination of meat by COLROS, highlighting the need for regular hazard analysis of critical meat-contamination points to prevent transmission of foodborne superbugs to potential meat consumers. Dismally, however, *mcr*-1-habouring *E. coli* have also been isolated from a human patient without a history of travel in Austria (*Hartl et al., 2017*), suggesting possible acquisition from livestock and community transmission. In 2015, Austria's livestock sector consumed 1,548 kg of COL, which is 300 times more than the five kg amount used in human medicine (*Kirchner et al., 2017*). Therefore, the selective pressure for mobilization of *mcr* in Austria might have originated from the livestock setting. However, it is worthy to note that *mcr*-1 was not detected in *E. coli* isolates from chicken meats collected from Austria in 2016 (*Zogg et al., 2016*).

In France, 26 strains (23 *E. coli* and 3 salmonellae) carrying *mcr*-1 and *mcr*-1-like genes were detected among 275 Enterobacteriaceae (9.4%) isolated from feces of poultry birds, chicken meat, ready-to-cook guinea fowl pie, chicken-farm boot (*Perrin-Guyomard et al., 2016*; *Webb et al., 2017*; *El Garch et al., 2018*), suggesting low circulation of *mcr*-1 among *Enterobacteriaceae* in the French poultry sector, that farmer's wears are potential vehicles for transporting COLROS from livestock farms to other places. The *mcr*-1 was on IncI, IncX4, and IncP plasmids in the *Salmonella* isolates (*Perrin-Guyomard et al., 2016*; *Webb et al., 2017*; *El Garch et al., 2018*), meaning that diverse plasmids evolved *mcr*-1 in France. This highlights the need for adequate biosecurity measures in livestock farms, especially disinfectant foot dips at farm entrances/exits to curtail the spread of MGHB from farm-to-farm and to public places. Lamentably, PMCR has already been disseminated into the French human-environmental ecosystem, possibly through livestock manure and/or livestock-to-human-to-human transmission (*Caspar et al., 2017*). A gene encoding AmpC was present in some of the *mcr*-1-harbouring *E. coli* isolates (*Perrin-Guyomard et al., 2016*), suggesting that *E. coli* coproducing MCR-1 and AmpC is present in the French poultry sector. Notably, in the NDARO database, a sequence of *mcr*-5.1 was detected in *S. enterica* isolated from boot used in a French chicken farm, suggesting that *mcr*-5 has been present in France.

The German poultry sector has been reported as a reservoir for MGHB (*Ewers et al., 2016*; *Falgenhauer et al., 2016*; *Irrgang et al., 2016*; *Zogg et al., 2016*; *El Garch et al., 2018*; *Eichhorn et al., 2018*; *Hornsey et al., 2019*; *Borowiak et al., 2020*). Reported consumption of COL in food animals in Germany was > 100 tonnes (*Webb et al., 2017*). Compared to 30 European countries, Germany had the sixth-highest polymyxin sale for food animals in 2016 (*Borowiak et al., 2020*). Thus, COL selective pressure has been exerted in the German livestock sector. A total of six hundred and eighty strains carrying various *mcr* genes (*mcr*-1–585 *E. coli* and 53 salmonellae, *mcr*-2–10 salmonellae, *mcr*-3–10 salmonellae and 1 *Aeromonas media*, *mcr*-5–19 salmonellae, *mcr*-4 and *mcr*-9–1 *Salmonella* each) were detected among 1,190 isolates (57.1%) recovered 2008–2018 from poultry birds (live and dead), meats and environment, including meats exported from Germany to Switzerland (*Ewers et al., 2016*; *Falgenhauer et al., 2016*; *Zogg et al., 2016*; *Borowiak et al., 2017*; *Eichhorn et al., 2018*; *Eichhorn et al., 2018*; *Hornsey et al., 2019*;

*Borowiak et al., 2020*). This further shows a high prevalence of *mcr*-harbouring organisms in German poultry since at least 12 years ago and that these organisms are causing difficult-to-treat diseases in the birds. The *mcr*-1 was on > 200 kb IncHI2 and IncX4 plasmids in some of the isolates (*Ewers et al., 2016*; *Hornsey et al., 2019*), while in others, it was also in the chromosome (flanked by IS*Apl1*) together with the ESBL gene (*bla*$_{CTX-M-15}$) flanked by IS*Ecp1* (*Falgenhauer et al., 2016*). The *mcr*-3 was a novel gene, named *mcr*-3.7 on 80 kb plasmid while the *mcr*-5 (a new gene) associated with transposon Tn3 as well as *mcr*-1 and *mcr*-3 were on ColE plasmid in more than half of the *mcr*-5-positive salmonellae (*Borowiak et al., 2017*; *Eichhorn et al., 2018*). This suggests that various genetic elements (transposons and common and uncommon plasmids) have circulated diverse *mcr* genes in Germany. These genes could persist in the farm through vertical transmission among clones. It also revealed that the German poultry sector constitutes a reservoir for the emergence of novel COL determinants. Two *S.* Paratyphi B var. Java possessed both *mcr*-1 and *mcr*-9 (*Borowiak et al., 2020*), suggesting that selective pressure by different antimicrobials could facilitate carriage of diverse *mcr* genes on a plasmid and that the German poultry industry constitutes a source for organisms capable of causing untreatable diseases. There was a chromate and ESBL gene in some of the *mcr*-5-harbouring salmonellae even in tigecycline-resistant isolates (*Borowiak et al., 2017*; *Borowiak et al., 2020*), 13 other resistance genes in five different antimicrobial families were in the *mcr*-3-positive *Aeromonas* (*Eichhorn et al., 2018*) whereas 18 extra resistance genes (including ESBL genes) were present in the *mcr*-1-positive isolates which were extensively diverse belonging to seven STs (dominated by ST156 and ST1431) (*Ewers et al., 2016*; *Falgenhauer et al., 2016*; *Zogg et al., 2016*; Horseny et al., 2019) (Table 2), including HiR pandemic ExPEC clones ST69, ST131, ST410, ST10 and ST58 (*Manges et al., 2019*). Thus indicating that multi to pandrug-resistance transmission among *Enterobacteriaceae* in Western Europe is clonally unrestricted. It also demonstrates that *Aeromonas* could transfer cocktails of multi-resistance determinants to other organisms. The heavy metal-resistant *Salmonella* coproducing MCR-5 and ESBL have been present in the German poultry environment, posing a significant public health risk. The chromate gene could code resistance against disinfectants, thereby causing a breach of biosecurity measures in livestock farms. Although the *mcr*-1 was located on a plasmid in the *E. coli* strains, conjugation was negative (*Hornsey et al., 2019*), further supporting that plasmidal location of *mcr* gene does not necessarily imply transferability. However, *mcr*-1-carrying *E. coli* has already disseminated into the human-environmental ecosystems in Germany (*Guenther et al., 2017*), possibly through contact with and/or consumption of contaminated livestock/livestock-related products. Nonetheless, the genome of *mcr*-3- and *mcr*-5-habouring strains conjugated with that of recipient organisms (*Borowiak et al., 2017*), implying that *mcr*-3 and *mcr*-5 could be transferred to other organisms.

Since *mcr*-positive isolates were from meat samples, it suggests that meat handlers and the meat-processing environment in Germany are potential sources for cross-contamination of meats with multi to pandrug resistant virulent *E. coli* clones, posing a grave danger to handlers of raw and/or undercooked meats. This calls for urgent attention since Germany is a prominent exporter of poultry meat in the EEA. It supports

the need for post-processing assessment of meat for COLROS/*mcr* genes and adequate meat cooking. Moreover, since *Aeromonas* is a known reservoir for *mcr*-3 and also a common inhabitant of aquatic system (from where *mcr*-3, *mcr*-4, and *mcr*-7 originated), avian and human gut (*Shen et al., 2020a*), the fish meal often used as a source of protein in livestock feed constitute a potential source of *mcr*-3-harbouring *Aeromonas.*

Furthermore, 85% of the *mcr*-positive salmonellae were susceptible to COL (MIC ≤ 2 mg/L); hence they were not considered earlier for PCR screening (*Borowiak et al., 2020*). This indicates that by using methods targeting a wide range of *mcr* genes, there is a high likelihood of estimating the actual prevalence of *mcr*-carrying isolates from an ecological niche. It also indicates that studies in which all the currently known *mcr* genes (*mcr*-1 to *mcr*-10) were not screened possibly underestimated the magnitude of PMCR. This notwithstanding, the salmonellae might also be harbouring yet unknown *mcr* genes, which further causes the spread of PMCR. Hence, there is a high likelihood that novel *mcr* genes will continue to emerge. Therefore, revision of the current ECV breakpoint for COL resistance, affordable rapid test kits capable of detecting all the currently known *mcr* genes and possibly the ones yet to emerge, and increased application of high throughput methods (such as whole-genome sequencing (WGS)) in surveillance of PMCR is much needed.

*OCEANIA*

It is worthy to note that none of 256 avian pathogenic *E. coli* (APEC) isolated from poultry birds during 2007–2016 in Australia harboured *mcr* gene (*Cummins et al., 2019*). But, in Indonesia, which is a transcontinental country in both Asia and Oceania, 13 *mcr*-1-carrying strains were detected among 58 COL-resistant pathogenic *E. coli* (22.4%) isolated in 2017 from all the critical points (chickens at farm, farm litter and drinking water, processing unit and chicken meats in restaurants) of poultry meat supply chain (*Palupi et al., 2019*), suggesting that MGHB has been present in poultry sector in Oceania since at least 3 years ago and that PMCR is transferred from the farm-to-plate. This highlights the need for adherence to basic infection prevention and control (IPC) practices such as hand hygiene by animal and meat handlers (slaughterhouse personnel) and food preparers and adequate cooking of food. Nevertheless, it is worthy to note that none of nine COL-resistant isolates from slaughtered chickens sampled (in 2016) in Australia harboured *mcr*-1 to *mcr*-5 (*Bean et al., 2020*). However, an *arnA*-like gene possibly conferred polymyxin resistance in the isolates.

*North America*

In the NDARO database, *mcr*-1.1 and 26 other resistance genes (including ESBL, fosfomycin, and PMQR genes) in four antimicrobial families were detected in 13 *E. coli* isolated 2018 from chickens in the USA. This means that the US poultry sector is a reservoir for multi-to extensively drug-resistant *E. coli.* Regrettably, organisms coproducing MCR-1 and ESBL have been detected in human isolates from the USA earlier than now (*McGann et al., 2016*), suggesting a possibility of cross-contamination from the human to poultry sector in the US. It is worthy to note that *mcr*-9 carried on a 260–340 kb

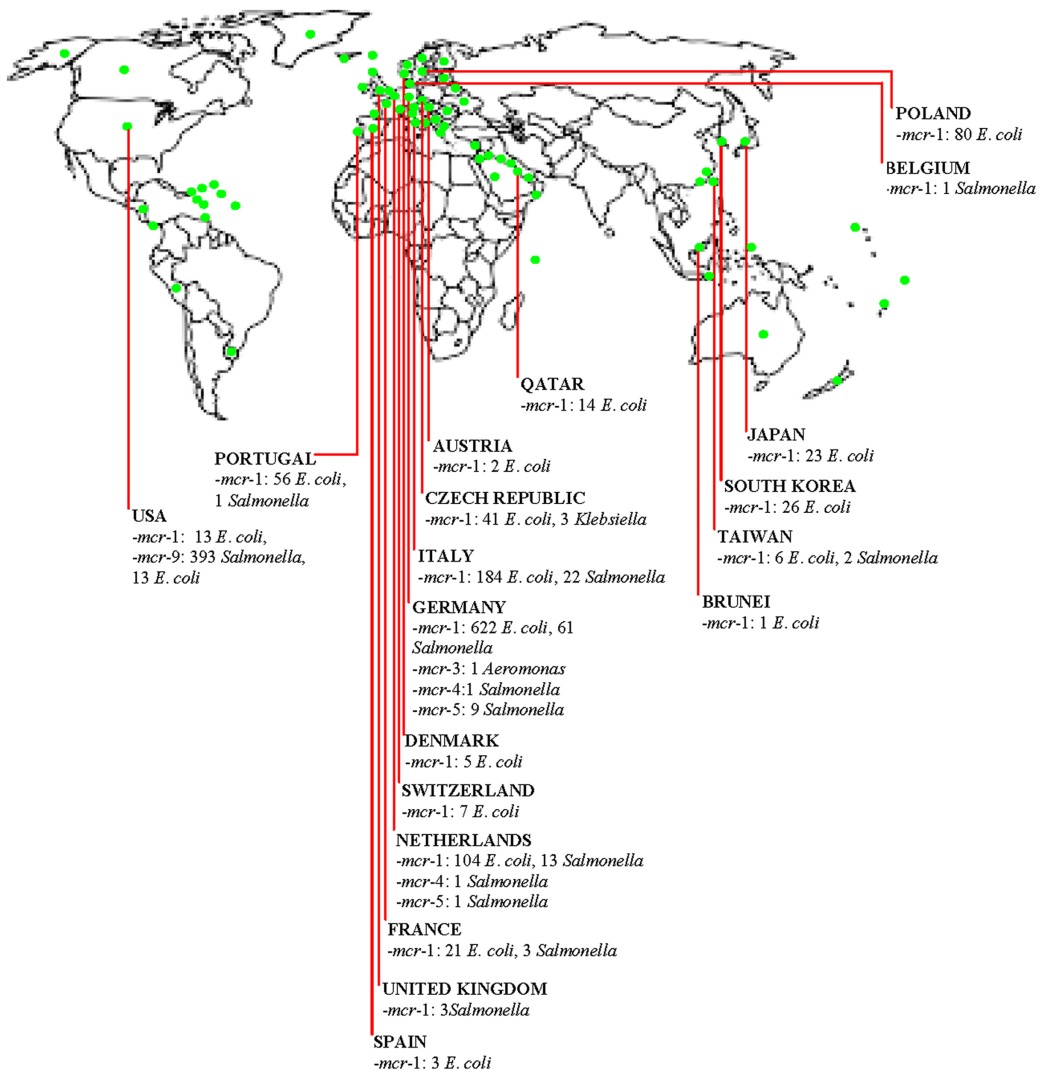

**Figure 1 Distribution of mobile colistin resistance (mcr) gene-carrying isolates in the poultry meat supply chain in high-income countries (HICs)/territories.** ● (Green dot) = High-income country, territory or area.                                        

IncHI plasmid, and chromosome was detected in 3,939 COL-susceptible (<1 µg/mL) *S. eneterica* (dominated by *S.* Saintpaul) and six *E. coli* isolated during 2002–2019 from poultry birds in the USA (*Tyson et al., 2020*). The *qseBC* two-component system, which is thought to induce *mcr*-9 expression (*Kieffer et al., 2019*), was downstream *mcr*-9 in some isolates, suggesting that the *qseBC* does not necessarily induce *mcr*-9 expression. Although the *mcr*-9 did not confer polymyxin resistance in all isolates and their transconjugants, there were six other resistance genes (including ESBL and heavy metal resistance genes) belonging to four antimicrobial families in the *E. coli* isolates. This poses a severe threat to animal and public health because organisms resistant to heavy metals and last-resort antibiotics could easily evade biosecurity measures (such as disinfectant foot dips, hand wash) in livestock farms. The fact that *mcr*-9 does not offer COL resistance highlights the need for AST of *mcr* gene-positive isolates (even after whole genome

sequencing) and to be cautious while interpreting the role of a novel COL resistance determinant.

*South America*

Out of the 2 HICs (Chile and Uruguay) in South America, only two studies have assessed the presence of *mcr* in the poultry sector in Chile (*Lapierre et al., 2020*) and Uruguay (*Coppola et al., 2020*). One strain positive for *mcr*-1 and fluoroquinolone resistance gene (*qnrB19*) was detected among 200 *E. coli* (0.5%) isolated from faecal samples of chickens in Uruguay (*Coppola et al., 2020*). This appeared indicative of the low circulation of *mcr*-1 in the Uruguayan poultry sector. Essentially, the use, sale and importation of COL for livestock has been banned in Uruguay since March 2019. Unfortunately, the *mcr*-1-harbouring *E. coli* has already disseminated into the human population (*Papa-Ezdra et al., 2020*). In Chile, 87 *Salmonella* Infantis were isolated from 361 broiler meat samples that were availed in supermarkets (*Lapierre et al., 2020*). However, none of the 25 isolates that exhibited COL resistance, were positive for *mcr* genes (*mcr*-1 to *mcr*-5).

## CONCLUDING REMARKS

The diversity of organisms such as *E. coli*, *Klebsiella*, *Salmonella*, and *Aeromonas* harbouring various *mcr* genes are widely spread in the poultry industry in HICs (Fig. 1). Additionally, *E. coli* was identified as the predominant organism spreading *mcr* genes in the poultry meat supply chain. Notably, the extensive use of COL/other antimicrobials in poultry, together with the importation of contaminated meats, are significant routes for development of PMCR in HICs. Clearly, in comparison to HICs where the use of COL in animal husbandry have been strictly regulated (such as Switzerland), there is higher prevalence of diverse MGHB in the poultry sector of HICs (such as Germany, Netherlands, Poland and so on) that relied or are relying on the importation of poultry birds/products and consumed high amount of COL in their livestock sector prior to restriction on non-therapeutic COL use. Thus, such countries constitute potential hotspots for the emergence and dissemination of diverse *mcr* genes across the globe. Enforcing a ban on the non-therapeutic use of COL in livestock could reduce the rate of development of MGHB. An epidemiological study from China proved that COL ban reduced the prevalence of MGHB (*Shen et al., 2020b*). Since non-therapeutic COL use has been restricted in most HICs, studies assessing the prevalence of COLROS after such interventions are needed to evaluate the magnitude of PMCR and enhance the control strategies for curbing the development and dissemination of COL resistance. Additionally, post-slaughter screening of meat for *mcr* gene could reduce the spread of PMCR by meat trade. Isolates of poultry origin in HICs contain *mcr* genes with diverse virulence and resistance (including AmpC, ESBL, carbapenemase, fosfomycin, and PMQR) genes. Thus, there are superbugs potentially causing difficult-to-treat diseases across both poultry farms and human populations. Some poultry isolates from HICs have acquired megaplasmid with numerous ARGs (some harbour ≥10 genes). The entry of these megaplasmids through farm-to-plate transmission into the human ecosystem could have a catastrophic impact on public health. Plasmids, including conjugative plasmids of different replicons and

incompatibility, truncated and composite transposons, especially IS*Apl1*, are drivers of PMCR in the poultry sector in HICs. These plasmids rapidly spread *mcr* genes by HGT to other organisms, having transferred to recipient organisms at a high frequency. Nonetheless, *mcr*-1, *mcr*-3, and *mcr*-9 have integrated into chromosomal DNA and non-conjugative plasmids in poultry strains from HICs. This ensures the vertical transfer and persistence of these genes among the clonal lineages. Besides, the *mcr* gene's transmission among poultry strains in HICs is clonally unrestricted, and diverse highly-virulent zoonotic pandemic and commensal clones of *Enterobacteriaceae* are circulating in the poultry industry in developed countries. Chromosomal mechanisms are also involved in COL resistance among isolates from poultry in developed countries.

Essentially, contact with poultry birds and manure, poultry farm workers/their wears and equipment is a potential route for acquiring PMCR. Consumption/handling of undercooked poultry meat is a putative route for colonization by MGHB. Poultry meat can be contaminated at the slaughterhouse by slaughterhouse personnel in HICs. Trade of poultry birds/meat and poultry-related products are routes for importing PMCR into HICs and other places. The use of insufficiently-treated/untreated poultry manure/slaughterhouse sewage as organic fertilizer is a potential source for disseminating PMCR into the human and environmental ecosystems in HICs. Evidently, by horizontal/lateral and vertical transfer, *mcr*-1, *mcr*-2, *mcr*-3, *mcr*-4, *mcr*-5, and *mcr*-9 have disseminated in the poultry sector in developed countries (Fig. 1). Farm-to-plate/farm-to-environmental transmission of PMCR from the poultry sector will increase in HICs (including countries yet with low PMCR prevalence) if efforts to curtail COL resistance in the poultry meat supply chain in developed and developing nations are not enhanced by the implementation of effective antimicrobial stewardship and use of antibiotic alternatives such as probiotics and antimicrobial peptides. This further highlights the need for the One Health approach. From all above, we are convinced that the policymakers, especially those involved in AMR, in addition to agricultural/poultry sector stakeholders-clinical microbiologists, farmers, veterinarians/veterinary officers, occupational health clinicians and related specialists, consumers/general public will find this current literature synthesis very useful.

### Funding
The publication is financed by University of South Africa, Johannesburg, Gauteng, South Africa. The funders had no role in study design, data collection and analysis, decision to publish, or preparation of the manuscript.

### Grant Disclosures
The following grant information was disclosed by the authors:
University of South Africa, Johannesburg, Gauteng, South Africa.

## Competing Interests

Charles Odilichukwu R. Okpala is an Academic Editor for PeerJ.

## Author Contributions

- Madubuike Umunna Anyanwu conceived and designed the study, analyzed the data, prepared figures and/or tables, authored or reviewed drafts of the paper, and approved the final draft.
- Ishmael Festus Jaja analyzed the data, prepared figures and/or tables, authored or reviewed drafts of the paper, and approved the final draft.
- Charles Odilichukwu R. Okpala analyzed the data, authored or reviewed drafts of the paper, and approved the final draft.
- Chinwe-Juliana Iwu Jaja analyzed the data, authored or reviewed drafts of the paper, and approved the final draft.
- James Wabwire Oguttu authored or reviewed drafts of the paper, and approved the final draft.
- Kennedy Foinkfu Chah conceived and designed the study, analyzed the data, authored or reviewed drafts of the paper, and approved the final draft.
- Vincent Shodeinde Shoyinka conceived and designed the study, analyzed the data, authored or reviewed drafts of the paper, and approved the final draft.

## Data Availability

This literature review does not involve raw data or code.

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
