# Peer review of "Potential sources and characteristic occurrence of mobile colistin resistance (mcr) gene-harbouring bacteria recovered from the poultry sector: a literature synthesis specific to high-income countries"

_PeerJ, doi:10.7717/peerj.11606_

## Round 0.1 · original submission · Major Revisions

Although quite an interesting and relevant topic, the current version is not clear, needs several grammatical and linguistic corrections, needs to be written precisely, and also to correct tables and figures. Please see the extensive review comments by reviewers, and try to incorporate them in the revision.

Reviewer 1 ·

Basic reporting

No comment

Experimental design

No comment

Validity of the findings

No comment

Additional comments

Comments_peerj2020_1
The authors review the mcr-harboring bacteria of poultry origin in high-income countries. The manuscript is very long but loose and difficult to read. The sequence of the context is organized disorderly. Numerous numbers (of isolates, date, …) listed in the Tables are inconsistent with those described in the text. It is recommended that authors check the tables and the text for errors and arrange the order of countries or regions in the tables to match the narrative order of the text. The manuscript should be shortened by half to make it more focused and concise.

Other comments:
1. Line 38: Since only very few studies characterized the virulence genes of isolates, it is suggested that the authors exclude this part (virulence) to make the review article more focused.
2. Line 145: spelling error “Enterobacteriacae”
3. Lines 59-60: should give a citation for this statement
4. Line 211: Ninety-nine “strains”… Here, “isolates” rather than “strains” is more appropriate. For many places in the text, isolate(s) instead of strain(s) is more proper.
5. Lines 211-212: “Salmonella enterica subspecies Typhimurium” The full name should be Salmonella enterica subspecies enterica serovar (or serotype) Typhimurium (abbreviation: S. Typhimurium)
6. Line 216: To replace “Salmonella enterica subspecies enterica Typhimurium” with the abbreviation “S. Typhimurium” and to use the abbreviation in the following text.
7. Lines 366-367 (and several places in the text): …In Portugal, 57 strains (56 E. coli and 1 Salmonella enterica Typhimurium) were detected among 242 Enterobacteriaceae (23.6%) … It is recommended moving the percentage (23.6%) to the position after the number (57)… as “In Portugal, 57 (23.6%) isolates (56 E. coli and 1 S. Typhimurium) were detected among 242 Enterobacteriaceae isolates …
8. Line 626: To replace “S. Paratyphi B dT+” with “ S. Paratyphi B var. Java” (in all the text and tables). D-tartaric acid “–“ is S. Partyphi B, d-tartaric acid “+” is S. Paratyphi B var. Java
9. Line 673: “APEC isolated” To spell out the full name of APEC as “avian pathogenic E. coli (APEC) isolates”
10. Line 684: “mcr-“ Is mcr-1?
11. Line 678 and the paragraph: mcr-9 has been shown not to offer resistance to colistin; it should not be important to this article and not be described as the first paragraph in the section of North America.
12. Line 728: … in the High-income countries (HICs)… The full name of HICs should not be re-written here again.
13. Line 735: both terms “farm-to-table” and “farm-to-plate (line 680)” are mentioned in the text. Should use only one?
14. Table 1: the column title “Source of isolate (Number of birds)” However, no “number” appears in this column. South Korea: mcr-1 is detected in 10 E. coli isolates but 12 STs are identified for the isolates. Taiwan: Abbreviation “S. Typhimurium” is recommended; 3 STs are listed but which one(s) are for the S. Typhimurium isolates and which one(s) are for the E. coli isolates? It is recommended that virulence genes should not be discussed in this review article.
15. Table 2: Netherlands: mcr-1 is detected in 14 E. coli and 13 Salmonella isolates and 10 STs are listed for the isolates; which STs are for the E. coli isolates, which one(s) are for the Salmonella isolates?

Reviewer 2 ·

Basic reporting

The review entitled "Potential sources and characteristic occurrence of mobile colistin resistance (mcr) gene-habouring bacteria recovered from the poultry sector: A literature synthesis specific to high-income countries" by Madubuike Umunna Anyanwu et al. aims to undestand the sources, prevalence, and genotypic characteristics of mcr-haroring bacteria in the poultry sector found in high-income countries. Overall, this review discusses the factor that could lead to a high prevalence of mcr-harboring bacteria and by comparing the countries between them show a general view of the current situation and action to reduce this prevalence. However, this review need to address a number of points in its current form. These are highlighted below.

Line 29: "Currently, 10 mcr genes (mcr-1 to mcr-10) are described". I sugget to change "are" by "have been".

Line 32: Escherichia coli, Klebsiella, Salmonella, and Aeromonas isolates. Klebsiella spp. or Klebsiella pneumoniae. The same for Salmonella.

Line 38: virulence factors. factors word should be added.

Line 70: "clinicians were forced to using COL for the treatment of". I sugget to change "using" by "use".

Line 80: "Plasmids are self-replicating DNA independent of chromosomes, and
80 they are highly mobile, carrying resistance and virulence-associated genes (VAGs)". This is not true at all, some plasmid are non-conjugative plasmid and need the machinary from another plasmid to be mobilized.

Line 112-117: "To reiterate the potential sources of plasmid-mediated colistin resistance (PMCR), particularly within the poultry sector, and to be specific, high-income countries (HICs) should be considered of very great importance. More importantly, to gather information about the characteristic occurrence of mcr gene-habouring bacteria (MGHB) recovered from the poultry sector specific to the high-income countries should also be considered of great importance". This part is repetitive and useless for the reader, it should be shorter. I suggest -> the potential source and the characteristic ocurrence of mcr-harboring bacteria (MGHB) particularly within the pultry sector specific to the high-income countries should be considered of great importance.

Line 189: "Nevertheless, some plasmids could increase in frequency and capture ARGs even in the absence of selective pressure (Lopatkin et al., 2016)".This is not correct and the reference used not say that the absence of exposition to antibiotics can increase the frequency of ARGs. I suggest to eliminate from the sentence -> "increase in frequency".

Line 259 and 262: I found reference missing at the end of the paper and names that were wrongly writed on the text. Please check again carefully all the references since it is a review and it is one the most important part due to the acquisition of the information it is from the reference. The same mistake were also found in Line 603,604,605,613 and 614,631,649.

Line 352: "transmission since the COL determinants were also detected". You should include resistance between COL and determinants.

Line 366: You should write -> 57 mcr-harboring strains.

Line 571: Sometimes the prevalence in percentage is given in comparison with all samples tested and in this part related to the COL-resistant isolates, creating the false impresion to the reader of a high prevalence of mcr-1. Considering the strains tested on the reference Allerberger et al. 2016 (164) and the mcr-1-positive isolates found. the prevalence is 1.8%.

Line 578: During all the text, you are using in excess the word "Unfortunately" that have sinonyms such as lamentably regrettably or dismally. By this way, the repetition could be avoided.

Line 649: Change mcr-1 by mcr-5.

Line 696: "However, the qseBC two-component system, which is thought to induce mcr-9 expression". Reference of the induction of mcr-9 expression should be included. DOI: 10.1128/AAC.00965-19.

Table 1: Remove "Number of birds" second column.

Table 2 (France): Instead of 24 is 275 in column 4. In column 5 is 3 and 23 instead of 2 and 21.

Table 2 (Austria): 168 instead of 4.

Experimental design

no comment

Validity of the findings

The conclusions should be more complete, I missed mainly a comparison in term of mcr genes prevalence between the different HICs countries and the factors that enhance this prevalence in comparison with other HICs such us, high level of food importation from developing countries, heavily use of COL in livestock, etc. This discussion between the different HIC countries would increased the quality of the conclusion.

Reviewer 3 ·

Basic reporting

no comment

Experimental design

no comment

Validity of the findings

no comment

Additional comments

Rapid dissemination of mobile colistin resistance (mcr) genes is particularly concerning and poses a major medical challenge. This manuscript described the dissemination of the mcr alleles(mcr-1 to mcr-10) among Enterobacteriaceae strains collected from food animals in high-income countries. The process of acquisition of mobile colistin resistance gene depends on various genomic background associated with insertion sequences. mcrs can be quickly transmitted among bacterial community through horizontal transfer. Also, the authors presented the other information of mcrs gene, including their prevalence, plasmids type and sequence type.

Limitations:
The authors collect information about all mcrs gene (mcr-1 to mcr-10) but did not listed the similarities and differences among these genes in different high-income countries. Overall, the logic of this manuscript was a little bit messy even though heading was used, which’s so hard to quickly get useful information without a clear structure. There are many grammatical mistakes needed to be corrected. Besides, the tables were not definite, terse, lacking clear, logical and comparative function.
I’m afraid this manuscript is more discursive than necessary and would benefit from some condensation. To be honest, I afraid that this manuscript is not well organized and would not be appropriate for PeerJ. It would also benefit from attention to the following points.

Detail points:
Line 72: It is not appropriate to use ccrB, it would be crrAB. Also, the related references should be used here.
Line 86: It would be “lipopolysaccharide of lipid A” rather than“lipid A lipopolysaccharide”.
Line 87-88: Authors should correct the describe of the mcr-1-mediated colistin mechanism.
Line185-188: It is not appropriate to say that China services as a source for dissemination of plasmid-mediated colistin resistance. Authors should be careful to use the word of causal relationship. Recent literatures in Lancet Infect Dis have been showed that the decreasing use of colistin in agriculture have had a significant effect on reducing the prevalence of mcr-1 in China.
Additionally, it would be better to list actions taken by authorities in response to the mcrs gene and summary the prevalence of mcrs gene after the intervention. For example, the European Medicines Agency updated advice on the use of colistin in European veterinary practices and recommended that colistin should be included in category 2 of Antimicrobial Advice ad hoc Expert Group.
For Figure1, the summary was partial, like mcr-1, mcr-3, and mcr-5 have been detected in E. coli derived from diseased pigs in Japan.
For the tables, authors should better highlight the important information.

---

## Round 0.2 · accepted · Accept

The authors revised their manuscript considering all suggestions, incorporating additional information, and it could be accepted in the current version.

Reviewer 2 ·

Basic reporting

no comment

Experimental design

no comment

Validity of the findings

no comment

Additional comments

no comment